# Characterization and Evaluation of 3D-Printed Connectors for Microfluidics

**DOI:** 10.3390/mi12080874

**Published:** 2021-07-26

**Authors:** Qianwen Xu, Jeffery C. C. Lo, Shiwei Ricky Lee

**Affiliations:** 1Smart Manufacturing Thrust, Systems Hub, The Hong Kong University of Science and Technology, Guangzhou 511458, China; qxuak@connect.ust.hk; 2HKUST Foshan Research Institute for Smart Manufacturing, The Hong Kong University of Science and Technology, Hong Kong 999077, China; jefflo@ust.hk; 3Electronic Packaging Laboratory, The Hong Kong University of Science and Technology, Hong Kong 999077, China; 4HKUST Shenzhen-Hong Kong Collaborative Innovation Research Institute, Shenzhen 518000, China; 5HKUST LED-FPD Technology R&D Center at Foshan, Foshan 528200, China

**Keywords:** 3D printing, microfluidics, interconnection

## Abstract

3D printing is regarded as a useful tool for the fabrication of microfluidic connectors to overcome the challenges of time consumption, clogging, poor alignment and bulky fixtures existing for current interconnections. 3D-printed connectors without any additional components can be directly printed to substrate with an orifice by UV-assisted coaxial printing. This paper further characterized and evaluated 3D-printed connectors fabricated by the proposed method. A process window with an operable combination of flow rates was identified. The outer flow rate could control the inner channel dimensions of 3D-printed connectors, which were expected to achieve less geometric mismatch of flow paths in microfluidic interfaces. The achieved smallest inner channel diameter was around 120 µm. Furthermore, the withstood pressure of 3D-printed connectors was evaluated to exceed 450 kPa, which could enable microfluidic chips to work at normal pressure.

## 1. Introduction

Microfluidics is the physics of manipulating fluids at a micro scale, typically the sub-millimeter scale [1]. It has been used in lab-on-a-chip (LOC) systems, micro-thermal management, inkjet printing technology and other potential fields. Among them, LOCs have promoted the evolution of portable and point-of-care diagnosis for human health. The fabrication of microfluidic devices attracts interests who come from both academic research and industrial companies. Various techniques have been proposed for microfluidic fabrication such as micromachining, molding, laser writing and additive manufacturing. Different types of microfluidic chips bring a challenge to the universally accepted microfluidic interface design.

Interconnection strategies are significant for microfluidic devices to allow a non-leaking, efficient and stable fluid pathway from the macro scale to the micro scale. Common microfluidic interface solutions can be divided into two main categories: (1) permanent gluing and sealing connections [2,3,4,5,6] and (2) reversible interconnections with fixtures [7,8,9,10,11]. Permanent gluing and sealing connections usually involve adhesives to bond tubing or fits onto chips, while reversible interconnections employ fixtures to introduce a compression force for tightness. Problems exist for each category, as in clogging or poor alignments for adhesives/gluing and cumbersome interfaces for reversible interconnections with fixtures.

With the coming of a new era in additive manufacturing [12,13,14,15,16], 3D printing provides convenience and design flexibility to the fabrication of microfluidic interconnections. These 3D-printed connectors can be either simultaneously printed fluid ports on 3D-printed microfluidic chips [17,18] or separated adapters fixed on microfluidic devices [19,20,21]. However, they are either limited to microfluidic types or difficult for high-density integration.

A general interconnection solution for most existing microfluidic devices is necessary to ease microfluidic fabrication. Thus, coaxial 3D-printed connectors without any additional components or adhesives have been reported by us before [22]. Previous works have successfully demonstrated the feasibility of coaxial 3D-printed connectors for microfluidic applications. Nevertheless, connector dimensions play an essential role in microfluidic operations for real applications, especially inner channel diameters. For example, geometry mismatch of flow channels in the interface will influence the flow path and result in dead volume issues or local pressure loss, especially for volume-sensitive analytical applications [23,24]. Therefore, controlling the flow path with less channel geometry mismatch inside the interface is crucial for microfluidic interconnections. In addition, the maximum working pressure of connectors is another significant consideration for their application. Examples of such claims can be found in chromatography, microreactors and sampling [25,26,27].

In this paper, we further discuss the parametric and statistical studies for 3D-printed microfluidic connector manufacturing. First, the process window for 3D-printed microfluidic connector manufacturing was investigated and concluded. Second, the effects of flow rates on inner channel dimensions were analyzed. Moreover, we also evaluated the hydrostatic strength of 3D-printed connectors.

## 2. Materials and Methods

### 2.1. Fabrication of 3D-Printed Microfluidic Connectors

A UV-assisted, coaxial 3D printing system for microfluidic connector manufacturing was developed to directly print connectors on the orifices of substrates. Herein, Figure 1a shows the experimental setup in the study. It consisted of a commercial dispenser (PACHDA, Shenzhen, China) with a movement stage, an assembled coaxial nozzle, two pressure-driven syringe pumps wherein stepper motors could control flow rates and a converged UV–LED light source for in–situ curing.

The coaxial nozzle replaced regular dispensing needles and was fixed on the robot arm. Figure 1b represents the design of the coaxial nozzle, which was assembled by two steel needles with other accessories. A 28 G (ID 0.17 mm, OD 0.35 mm) inner steel nozzle was chosen for water extrusion, and an 18 G (ID 0.91 mm, OD 1.26 mm) outer steel nozzle was employed for UV adhesive (LOCTITE 3491, Henkel AG & Company, Düsseldorf, Germany) extrusion. As discussed in previous works [22], it takes at least 6 s for LOCTITE 3491 to stabilize its structure under the shine of a converged UV–LED light source. Thus, we introduced a transparent jacket (ID 1 mm, OD 2 mm, length 25 mm) to assist the pre-curing step for UV adhesive before its deposition. Otherwise, the 3D-printed connector would easily collapse. The material of the transparent jacket was chosen as silicone due to its hydrophobic property, which resulted in lower friction force during printing. In addition, as the UV radiation length utilized for in situ curing was 6 mm, the printing speed was set to 1 mm/s to ensure enough UV exposure.

Figure 1c depicts the printing strategy for microfluidic connector fabrication. Two fluids were extruded simultaneously from the coaxial nozzle during printing and then flowed together to pass through the transparent jacket. If they were not mixing, the core liquid could present different coaxial flow patterns such as continuous jetting or dripping (pinching off to droplets). When the coaxial flow pattern maintained a continuous-jetting state, hollow connectors printed on substrates with orifices became possible if we selected the inner fluid (water) for the sacrificial fluid and outer fluids (UV adhesive) for shell formation under enough UV exposure. Polymethyl methacrylate (PMMA) substrates with orifices of 2.5 mm have been used in subsequent studies. After printing, a post-curing step is necessary for 3D-printed connectors. Here, we used a high-intensity UV mercury lamp for post curing for 2 min.

### 2.2. Evaluation of 3D-Printed Microfluidic Connectors

Pressure consideration plays a vital role in the design and manufacturing of 3D-printed microfluidic connectors. To estimate the maximum pressure that 3D-printed connectors could withstand with no leaks, a pressure test was performed. According to the standard of the ASME B31 pressure piping code [28], pressure tests can be done either with liquid, usually water, or with gas, usually compressed air or dry nitrogen. Furthermore, the pressure drop of the system or bubble emission when the component emerges in water is implemented to identify leaks. Typically, a leak test must be maintained for at least 10 min or for the length of time that it takes to inspect leaks visually.

In this study, we selected compressed air for the pressure test. Figure 2 illustrates the schematic demonstration of the pressure test for 3D-printed connectors on PMMA substrate. In testing, a compressed air source, a control valve, a pressure regulator, a pressure gauge and the 3D-printed connector were connected by tubing and adapters. The control valve was utilized to input or isolate the compressed air, while a pressure regulator was used for pressure control. The pressure gauge (Asmik, Hangzhou, China) was employed for pressure measurements with a range and an accuracy of 0–1 MPa and 0.5%, respectively.

Moreover, UV-cured adhesive was applied to seal the bottom of the connector and the connector’s interface to make a dead end. Double-sided tape was used to avoid additional adhesive flow into the connector channel. Eventually, the 3D-printed connector with a dead end emerged in a water tank for leak detection.

### 2.3. Dimensional Characterization

To determine the geometry of 3D-printed connectors, we further characterized their dimensions under the microscope such as outer diameter, inner channel diameter and inner channel waviness. Due to the printing strategy, 3D-printed connectors were constrained in the transparent jacket for solidification and then deposited on the substrate. Therefore, the outer diameters of 3D-printed connectors were fixed to 1 mm, which was conformal with the inner diameter of the transparent jacket. As for the inner channel diameter and waviness, we measured the inner channel diameter of the 3D-printed connectors at multiple locations with a spacing of 1 mm. Subsequently, the inner channel diameter was averaged, and its standard deviation represented the inner channel waviness.

### 2.4. Statistical Analysis

For dimensional analysis of 3D-printed connectors, the statistical significance for experimental data was calculated by one-way analysis of variance (ANOVA) with a significance level of *p* < 0.05 in Origin 2018 (OriginLab, Northampton, MA, USA).

## 3. Results and Discussion

### 3.1. Process Window for 3D-Printed Connector Manufacturing

With this particular coaxial nozzle design, a fixed printing speed and two selected fluids, flow rates were controllable parameters (Q_i_: inner flow rate, Q_o_: outer flow rate) determining the printing results. Flow rates affected the coaxial flow patterns in the coaxial flow system and thereby influenced the inner channel dimensions of the 3D-printed connectors. A parametric study was performed with variable flow rates for both inner fluid and outer fluid to obtain the process window of the successful printing of connectors. All results under different combinations of flow rates from experimental works can be categorized into blocking, feeding limitation, successful printing and thin shell, as illustrated in Figure 3.

For the combination of small Q_i_ and large Q_o_, the inner channels of printing results tended to block due to insufficient water extrusion for the sacrificial. However, as Q_o_ was small, and Q_i_ was large, the printing results showed a thin shell structure (see Example 2 in Figure 3). Due to insufficient extrusion of UV adhesives, the wall of the 3D-printed connector was too thin to work efficiently for microfluidic interconnections. Sufficient water extrusion and UV adhesive extrusion were required for successful printing, and they showed a competitive relationship. Thereby, well-shaped connectors (see Example 1 in Figure 3) could be fabricated with combinations of inner flow rate and outer flow rate, depicted in blue shadow. Furthermore, the syringe drivers used in the study had feeding limitations with relatively large flow rates. It was observed that syringe drivers could not consistently drive the syringe pumps when the outer flow rate reached 6 mL/min and the inner flow rate approached 8 mL/min.

Figure 4 shows the microscopic inspection and dimension measurements of a successful 3D-printed connector at Q_i_ of 5 mL/min and Q_o_ of 3 mL/min. Black ink was used to fill the tube for the ease of inspection. The length of 3D-printed connector was 5 mm, while the outer diameter was around 1 mm. The outer surface was quite smooth due to the constriction of the transparent jacket. The inner channel diameter was averaged to 155 µm, while the inner channel waviness was calculated to 11 µm.

To summarize, the obtained process window helped choose the appropriate combinations of flow rates to manufacture 3D-printed connectors. On the other hand, by comparing the coaxial flow behavior region under non-UV exposure in previous works [22] and the process window of connector printing under UV exposure, we found UV exposure conditions impacted the coaxial flow status. The reason is that the material viscosity of UV adhesives dynamically changes under UV exposure.

### 3.2. Effect of Flow Rate on Connector Dimensions

Inner channel dimension is the most significant parameter for the operation of 3D-printed microfluidic connectors. It impacts the flow path geometry, working pressure, flow velocities and other fluid parameters. Thus, controlling the inner channel size is very important for the fabrication of microfluidic connectors. In the UV-assisted coaxial 3D printing process, the controllable parameters for the inner channel dimension are the flow rates of two fluids, the nozzle dimension and UV intensity. In this study, the nozzle dimension and UV intensity were fixed. Thus, we investigated the effect of flow rates on the inner channel dimension of 3D-printed connectors.

Figure 5a represents the influence of the outer flow rate on the inner channel dimension. In experiments, the outer flow rate was changeable, while the inner flow rate was fixed to 5 mL/min. According to the results, the first three groups (Q_o_ = 2, 3, 4 mL/min) had significant differences from each other. Furthermore, as the outer flow rate increased, the inner channel diameter decreased, but the inner channel waviness did not represent an apparent tendency. However, the last group (Q_o_ = 5 mL/min) did not represent a significant difference with the third group (Q_o_ = 4 mL/min), as it reached the process window’s boundary.

Figure 5b describes the influence of the inner flow rate on the inner channel dimension. The inner flow rate was adjustable with a constant outer flow rate at 2 mL/min. Significance analysis was also conducted for the experimental data. However, the four groups (Q_i_ = 2, 4, 6, 8 mL/min) were not significantly different. The results claimed that the inner flow rate did not affect the inner channel diameter and waviness with a distinct increasing or decreasing trend.

To sum up, two flow rates are essential process parameters for the successful printing of 3D-printed connectors. The outer flow rate can adjust the inner channel diameter of 3D-printed connectors, whereas as the outer flow rate increases, the inner channel diameter decreases. The obtained smallest channel diameter of 3D-printed connectors was around 120 µm.

### 3.3. Hydrostatic Strength

To estimate the maximum working pressure that 3D-printed connectors could withstand, a pressure test was performed accordingly. The capability of the pressure test setup was identified to 450 kPa in preliminary experiments. Next, several samples (ID 170 µm, OD 1 mm, length 5 mm) were prepared for the pressure testing with a sample size of 8. The pressure applied to 3D-printed connectors was adjusted from 50 kPa to 450 kPa with intervals of 50 kPa. At each load, the test system was maintained for 10 min to observe any bubble emissions or pressure drop after being isolated from the compressed air source. Figure 6 shows the typical pressure test results for 3D-printed connectors, confirming no bubble emission or pressure drop in the testing. All samples were identified with no leaks when the loading pressure reached 450 kPa. Thus, it was concluded that the 3D-printed connectors on PMMA substrates could withstand pressures exceeding 450 kPa.

### 3.4. Benchmarking with Other Microfluidic Interconnections

3D-printed connectors in this study were benchmarked with other solutions for microfluidic interconnections. Various aspects needed to be considered involving the target material, interconnect density and maximum working pressure. Table 1 emphasizes the superiority of this work for microfluidic interconnections. First, the proposed 3D-printed connectors are compatible with different microfluidic types with a diversity of substrate materials due to the excellent adhesion performance of UV adhesives. In addition, the inner channel dimension is expected to be altered to fit target microfluidic channels and thereby eliminates the geometry mismatches of flow paths.

Second, the proposed 3D-printed connectors can enable high interconnect density by eliminating any fixtures or auxiliary structures used. Last but not least, 3D-printed connectors can withstand normal pressure for microfluidic chips. Therefore, the 3D-printed connector is expected to work efficiently and be reusable for microfluidic operations. The chemical compatibility of 3D-printed connectors is related to the properties of the material chosen for printing, which was acrylate in this study. Different UV curable polymers can also be utilized to fabricate printed connectors according to their particular working requirements.

## 4. Conclusions

Packaging strategies for microfluidic interfaces attract a lot of interest from both academic research and industrial companies. However, there are no universally accepted approaches, and only a few interconnection standards exist. To ease the microfluidic interconnection, we developed a general solution for microfluidic connectors through UV-assisted coaxial printing. In this work, the process window with an operable combination of inner flow rate and outer flow rate was investigated to achieve successful printing without blocking or thin shell structure. The effect of flow rate on connector dimensions was also identified. The increased outer flow rate represented a decreasing trend with inner channel diameter, but the inner flow rate did not affect the inner channel dimension with an increasing or decreasing tendency.

Obviously, the inner channel dimension of 3D-printed connectors can be altered to fit target microfluidic channels and thereby eliminate the geometry mismatch of the flow path. The obtained smallest inner channel diameter was ~120 µm. In addition, the maximum working pressure of 3D-printed connectors was estimated to exceed 450 kPa, which is accepted by microfluidic chips working under normal conditions. By benchmarking with other microfluidic interconnections, 3D-printed connectors have good compatibility with microfluidic chip types and the potential for high-density integration. These advantages make 3D-printed connectors promising for microfluidic applications.

## Figures and Tables

**Figure 1 micromachines-12-00874-f001:**
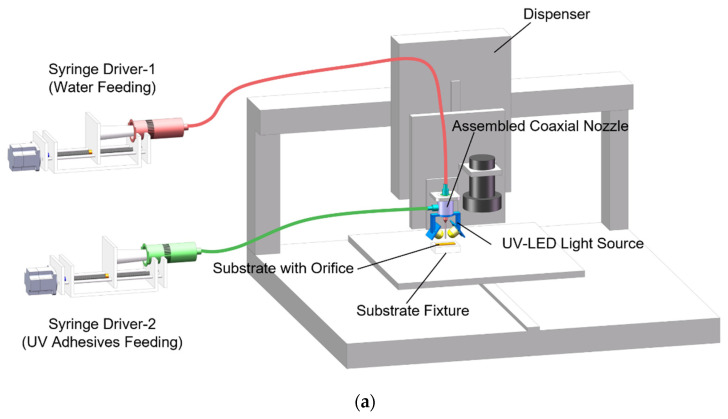
Experimental procedure for the fabrication of 3D-printed microfluidic connectors: (**a**) UV-assisted, coaxial 3D printing system; (**b**) design of the assembled coaxial nozzle; (**c**) flowchart of the printing strategy.

**Figure 2 micromachines-12-00874-f002:**
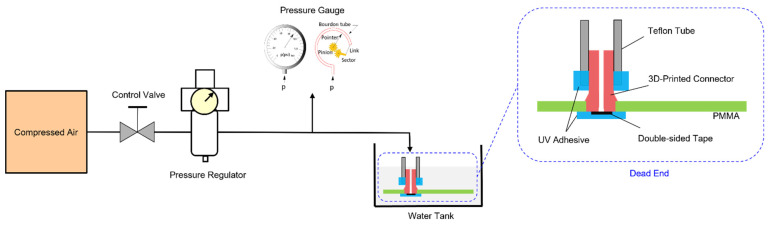
Schematic of pressure test for the 3D-printed connector on PMMA substrate.

**Figure 3 micromachines-12-00874-f003:**
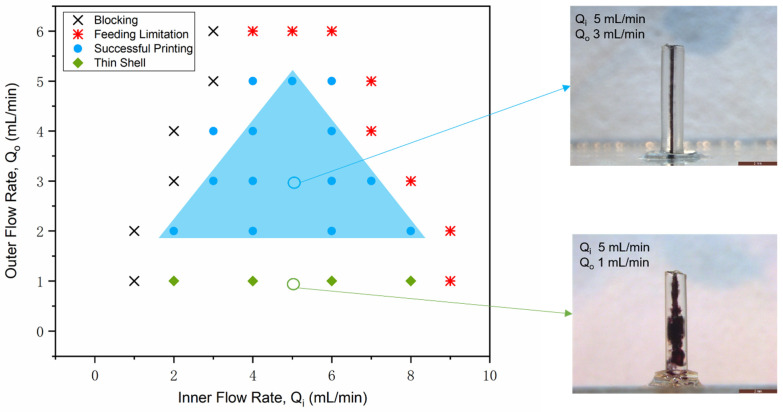
Process window for 3D-printed connector manufacturing, in which two typical examples of 3D-printed connectors are presented (Example 1: Q_i_ 5 mL/min, Q_o_ 3 mL/min inside the process window and Example 2: Q_i_ 5 mL/min, Q_o_ 1 mL/min outside the process window).

**Figure 4 micromachines-12-00874-f004:**
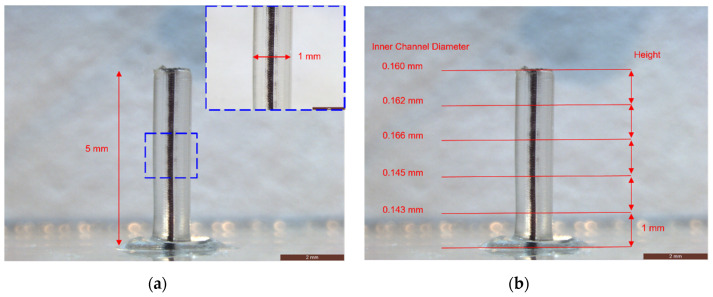
Dimensional characterization of 3D-printed connectors under the microscope: (**a**) the measurement of outer diameter and height; (**b**) the measurement of inner channel dimension.

**Figure 5 micromachines-12-00874-f005:**
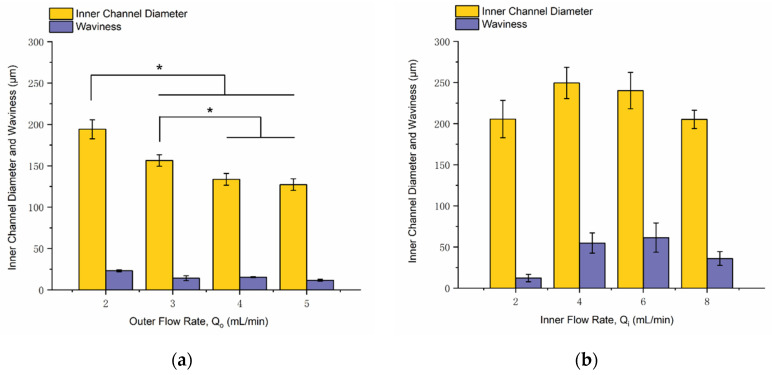
Effect of flow rate on channel dimensions of 3D-printed connectors: (**a**) the effect of outer flow rate under the same inner flow rate of 5 mL/min; (**b**) the effect of inner flow rate under the same outer flow rate of 2 mL/min (*n* = 3, single asterisk (*) indicates significant differences between groups (*p* < 0.05)).

**Figure 6 micromachines-12-00874-f006:**
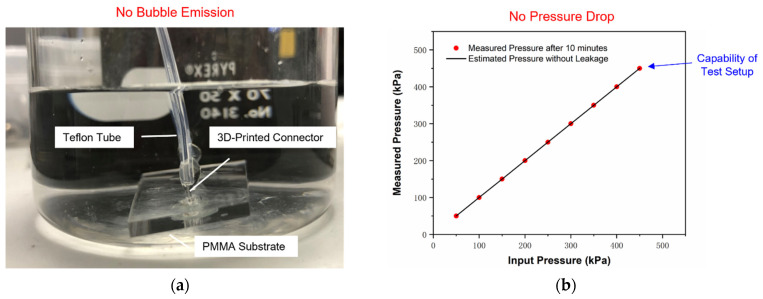
Typical results of the pressure test for 3D-printed connectors: (**a**) no bubble emission and (**b**) no pressure drop after 10 min at each load when the input pressure was adjusted from 50 kPa to 450 kPa (capability of test setup).

**Table 1 micromachines-12-00874-t001:** Benchmarking of 3D-printed connectors in this work with other microfluidic interconnections.

Strategy	Solution	Target Material	Interconnect Density	Maximum Working Pressure
Permanent Gluing and Sealing Connections	Adhesive-based gluing	All	High	190 kPa [4]
	Solder-based connections	Si	High	20 MPa [5]
	Ultrasonic welded connectors	Thermoplastic	Moderate	600 kPa [6]
Reversible and Fixing Interconnections	Compression sealing	PDMS	High	N/A [7]
	Self-aligning interconnects	Glass/plastic	High	3 MPa [8]
	Heat-shrink tubing sleeves	Si	High	200 kPa [9]
	Magnetic connectors	All	Low	500 kPa [29]
3D-Printed Connections	3D-printed fluidic ports	3D-printed microfluidics	High	N/A [17]
	3D-printed clamping interconnects	PDMS	Low	400 kPa [21]
	This work	All	High	≥450 kPa

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
