# Peer review of "Characterization and Evaluation of 3D-Printed Connectors for Microfluidics"

_micromachines, 2021, doi:10.3390/mi12080874_

Round 1

Reviewer 1 Report

Dear authors

The paper is quite good written but in my opinion its scientific level for this journal is too low. 

What are the common rules for assessing the accuracy of 3D printing fit, and why did the authors choose to compare the inner contact area and the inner gap?

No structural analysis and no assessment of the homogeneity of the printed materials.

There are no studies of the topography of the printed surfaces - what can have impact for microfluidics

No strength tests of the printed element what can confirm the transfer of high pressures of 450 kPa - can be added 

Conclusions should refer to the specific results of the research conducted and described in the article.

Reviewer 2 Report

Thank you for your submission. Your manuscript was reviewed. This paper is useful to the readership. The paper is well written. The authors conclusions are logical, sound and support the data. The references are current and appropriate. My only concern is statistical analysis. The sample size in the study was small but parametric test was used.

Reviewer 3 Report

Dear authors:

Please consider my comments below for your reference:

  1. 3D-printing of microfluidic devices is interesting. There are some good articles that might be useful to enhance the introduction section of the paper.

[1] 3D printing of metallic microstructured mould using selective laser melting for injection moulding of plastic microfluidic devices.

[2] Prototyping and Production of Polymeric Microfluidic Chip.

  1. In lines 68 and134 “to direct print a connector on the orifice of substrates” and “Flow rates were affecting the coaxial-flow patterns in the coaxial flow system and thereby influencing the printing results of microfluidic connectors”, please check the grammas, and go through the manuscript with correction of tense and symbols.
  2. Please check the font size of all figures, keeping them consistent with texts.
  3. In line 137, “eventually represent impacts on the printing results”, it is hard to understand.
  4. The same, in lines141-144, please check grammar.
  5. The heading of “3.2. Dimensional Characterization” can consider to be moved to material and methods section.
  6. For studying of the influence of the outer flow rate on the inner channel dimension, the results concluded that the out-flow rate of 5 mL/min reached the boundary of process window. How about the 6 mL/min?
  7. In section 3.3, “However, the result claims that the inner flow rate does not affect the inner channel diameter and waviness with a distinct trend, which may result from the impacts of UV radiation during printing”. However, it can be seen from Figure 5(b) that it do really have impact, can you check this and interpret it?
  8. The discussion is severely lack for the experimental results. Present manuscript is more like a technical report that is not sufficient to publish for scientific communication.
  9. The starting of the conclusion is overlap with introduction section, please rewrite the conclusion with key findings of the study and indicate their significance.
  10. The withstand pressure of 450KPa is accepted as a normal pressure, and can not be treated as high pressure for microfluidics. 

Author Response

Thank you!

Round 2

Reviewer 1 Report

The corrections introduced and the authors' explanations are sufficient.